# Assessment of Various Density Functional Theory Methods for Finding Accurate Structures of Actinide Complexes

**DOI:** 10.3390/molecules27051500

**Published:** 2022-02-23

**Authors:** Youngjin Kwon, Hee-Kyung Kim, Keunhong Jeong

**Affiliations:** 1Department of Mechanical System Engineering, Korea Military Academy, Seoul 01805, Korea; kyjchonje@kaist.ac.kr; 2Nuclear Chemistry Research Team, Korea Atomic Energy Research Institute, Daejeon 34057, Korea; hkkim@kaeri.re.kr; 3Department of Chemistry, Korea Military Academy, Seoul 01805, Korea

**Keywords:** DFT, actinides, americium (III) hexachloride, uranium hexafluoride, uranyl complex

## Abstract

Density functional theory (DFT) is a widely used computational method for predicting the physical and chemical properties of metals and organometals. As the number of electrons and orbitals in an atom increases, DFT calculations for actinide complexes become more demanding due to increased complexity. Moreover, reasonable levels of theory for calculating the structures of actinide complexes are not extensively studied. In this study, 38 calculations, based on various combinations, were performed on molecules containing two representative actinides to determine the optimal combination for predicting the geometries of actinide complexes. Among the 38 calculations, four optimal combinations were identified and compared with experimental data. The optimal combinations were applied to a more complicated and practical actinide compound, the uranyl complex (UO_2_(2,2′-(1E,1′E)-(2,2-dimethylpropane-1,3-dyl)bis(azanylylidene)(CH_3_OH)), for further confirmation. The corresponding optimal calculation combination provides a reasonable level of theory for accurately optimizing the structure of actinide complexes using DFT.

## 1. Introduction

Density functional theory (DFT) is the most widely used method for predicting the properties of molecules. The reliability of DFT results has led to increased applications in the fields of chemistry and materials science [1,2,3]. DFT studies have been essential for understanding rapid reaction processes [4] and have been used to calculate the electronic structure of molecules primarily composed of organic materials or molecules containing transition metals [5,6]. For the latter, it is not possible to include a multi-reference character in a method based on a single-configuration approach to represent an almost degenerate electron state. However, recent DFT studies have overcome this problem and afforded precise results on organometallic or metal clusters [6,7,8,9]. Nevertheless, DFT-derived information on molecules consisting of 92 or more electrons, such as actinides, is lacking [10,11,12] due to the high radioactivity of such molecules that must be handled in safe and appropriately designed control facilities [13,14]. Furthermore, the actinide orbital interaction model is challenging to calculate owing to the emergence of spin-orbit coupling, many-electron counts, and 5f and 6d bonding orbitals [15]. DFT studies have been used as an effective way to analyze the spectrum and geometry of this interaction model [16].

DFT has been successfully employed using various functional and basis set options for the molecular modeling and analysis of actinides. In many studies on actinides, methods like B3LYP, BP86, and PBE have been used. However, combinations of these methods have not been studied [9,17,18,19]. Studies on organic materials and comparisons of the methodologies used for transition metals, such as Ni, Fe, La, and Gd, have been reported. However, similar comparisons for heavy metals like actinium are insufficient [8,20,21]. Moreover, actinide materials have been studied using DFT. However, these studies do not include combinations of methods [22]. Therefore, in this study, we compared each method using relatively simple structures, americium(III) hexachloride (AmCl_6_^3−^) and uranium hexafluoride (UF_6_), and verified that the selected methods could be used to analyze more complicated structures like the uranyl complex (UO_2_(L)(MeOH), where L = (2,2′-(1E,1′E)-(2,2-dimethylpropane-1,3-dyl)bis(azanylylidene) [9,16,23,24], as shown in Figure 1. Our theory was used to describe actinide chemistry to demonstrate the reliability of the results.

## 2. Results and Discussion

The bond distances of UF_6_ and AmCl_6_^3−^ were calculated using 38 different theoretical combinations and are shown in Figure 2 and Figure 3, respectively. The complex molecular structure of UO_2_(L)(MeOH) was calculated based on the three most accurate calculation methods used to calculate the structures of AmCl_6_^3−^ and UF_6_. The calculation methods used in this study optimized the geometries of atoms in the molecule and predicted the structure closest to that obtained from the experimental results. We represent the mean absolute deviation (MAD) between the experimental and calculated values instead of plotting the length of all calculated bonds.

### 2.1. Uranium Hexafluoride

UF_6_ is a widely known molecular compound used as a key ingredient in the enrichment of natural uranium [23]. Studies have been conducted based on the mean distance determined from automatic neutron diffractometry, infrared and Raman spectra, electron diffraction, and quantum calculations [18,21,24,25]. The mean bond lengths of the molecular structures optimized using 38 DFT calculation combinations are shown in Figure 2. The MAD calculations showed deviations ranging between 0.0001 Å and 0.04 Å. (see Appendix A).

### 2.2. Americium (III) Hexachloride

The structure of hexahedral AmCl_6_^3−^ was studied using DFT calculations and single-crystal X-ray diffraction (SCXD) [9]. The optimal distances were calculated using the combinations of DFT calculations used for UF_6_. A comparative analysis with reported experimental values is shown in Figure 3. PBE0/6-31G(d) and PBE0/6-31+G(d) were omitted because they did not converge despite extended time with an additional grid and a maximum cycle for the XQC algorithm. The MAD of the average bond length was between 0.06 Å and 0.15 Å.

### 2.3. Uranyl Complex (UO_2_(L)(MeOH))

Despite the structural deviation between UF_6_ and AmCl_6_^3−^ due to the charge difference and different configurations of valence electrons, which results in different bonding properties, the most accurate structures were the same: N12/6-31G(d), B3P86/6-31G(d), M06/6-31G(d), and B3PW91/6-31G(d). The additional diffuse function did not provide a more accurate optimized structure for AmCl_6_^3−^ or UF_6_. Despite the contentious relationship between covalency in f-element systems and structures, both octahedral structures play the same role in the bonding model [26]. Therefore, the increased covalent bonding characteristics play a major role in the interaction. For future applications, the analysis was conducted on a uranyl complex (UO_2_(L)(MeOH)) to verify the accuracy of three of the four optimized calculations (B3P86/6-31G(d), M06/6-31G(d), and B3PW91/6-31G(d)) when applied to larger, more complicated molecular structures [17]. N12/6-31G(d) was omitted because it could not be optimized despite our best efforts. The MAD of the lengths and angles between molecules was measured, and the results were compared, as shown in Figure 4 and Table 1. The results obtained by all these methods agreed with experimental data within 0.05 Å in length and 1.5° in bonding angle, with small deviations between methods, confirming that our systemic method for finding the optimal level of theory for calculating the structure of actinide complexes works well. B3PW91/6-31G(d) exhibited the smallest MAD among the three calculation combinations and accurately predicted the structure, in line with the experimental data, with length and angle deviations of less than 0.04 Å and 1.4°, respectively. Individual bond length and bond angle comparisons between each calculated result and the experimental result were carried out, showing that B3PW91 was the most accurate method for the experimental structure.

## 3. Computational Methods

All calculations were performed using the Gaussian09 software package [27]. AmCl_6_^3−^ and UF_6_ were optimized with frequency calculations using 19 functionals with two basis sets, resulting in 38 possible combinations of the levels of theory. After examining each frequency without an imaginary number, the average length of each bond was compared to the experimental molecular length reported in the literature [9,23]. We selected DFT calculations based on recent theoretical studies on actinide chemistry: B3LYP, BLYP, BP86, BPW91, PBE, N12, M06L, M11L, MN12L, TPSS, B3P86, B3PW91, PBE0, M06, TPSSh, wB97X, M11, N12SX, MN12SX, and the basis set for H, O, C, N, F, and Cl with 6-31G(d) to obtain relatively light complexity and accurate theoretical results [28,29,30,31]. Ligands with anions bind loosely on the actinides, and diffuse function addition may be better for precise descriptions of the actinide ligands’ complex structures. Therefore, 6-31+G(d) was used for theoretical calculations [8,21,32,33]. The ECP60MWB relativistic effective core potential and the associated basis set developed by the Stuttgart–Cologne group were selected to describe americium and uranium [29]. Several combinations of calculations were applied to UO**_2_**(L)(MeOH), and the results were compared to prove its accuracy [17]. The developed DFT method is one of the most accurate ones for computing the electronic structure of solids [34,35,36,37] and will help study actinide materials.

## 4. Conclusions

The properties of actinide complexes are difficult to optimize using DFT due to unreliable theoretical methods for compounds containing more than 92 electrons. Therefore, a systematic study of methods for obtaining the theoretical structures of actinide complexes is significant for future research on actinide chemistry. This study does not contain the full relativistic effect in actinides. However, a scalar-relativistic effect is considered that can be applied to future studies. Here, two representative structures of actinide complexes, UF_6_ and AmCl_6_^3−^, were used to determine the best DFT functions. From 38 computational combinations, the optimal combinations were evaluated by optimizing the geometries of the molecules and comparing the calculated results with the experimental values. The four most promising levels of theory for the two actinides were obtained and applied to UO_2_(L)(MeOH) to confirm the accuracy of the optimal computational methods when applied to large and complex molecular structures. Finally, the B3PW91/6-31G(d) calculation yielded structures closest to the predicted actinide structures, providing a standard for establishing the level of theory of actinide complexes in the future.

## Figures and Tables

**Figure 1 molecules-27-01500-f001:**
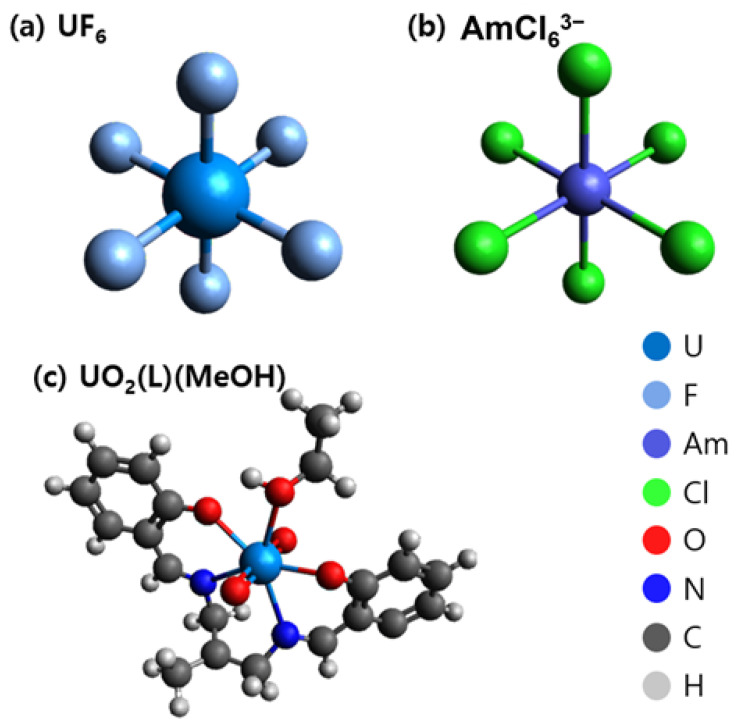
Molecular structure of (**a**) UF_6_, (**b**) AmCl_6_^3−^, and (**c**) UO_2_(L)(MeOH) [9,17,23,24].

**Figure 2 molecules-27-01500-f002:**
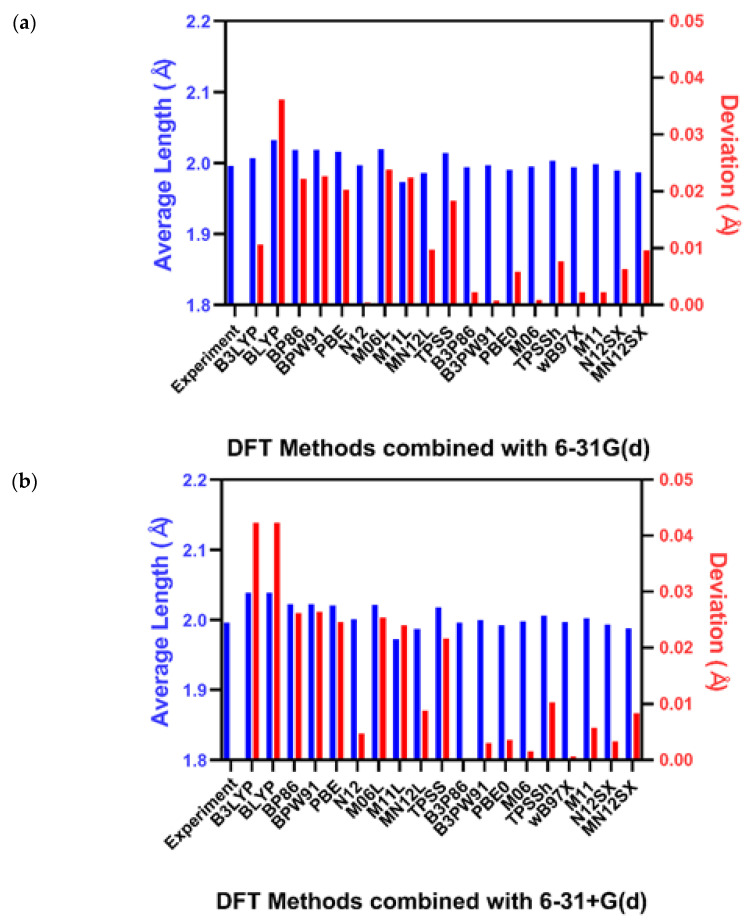
Average length of U-F bonds in the optimized structure with (**a**) 6-31G(d) and (**b**) 6-31+G(d) [23].

**Figure 3 molecules-27-01500-f003:**
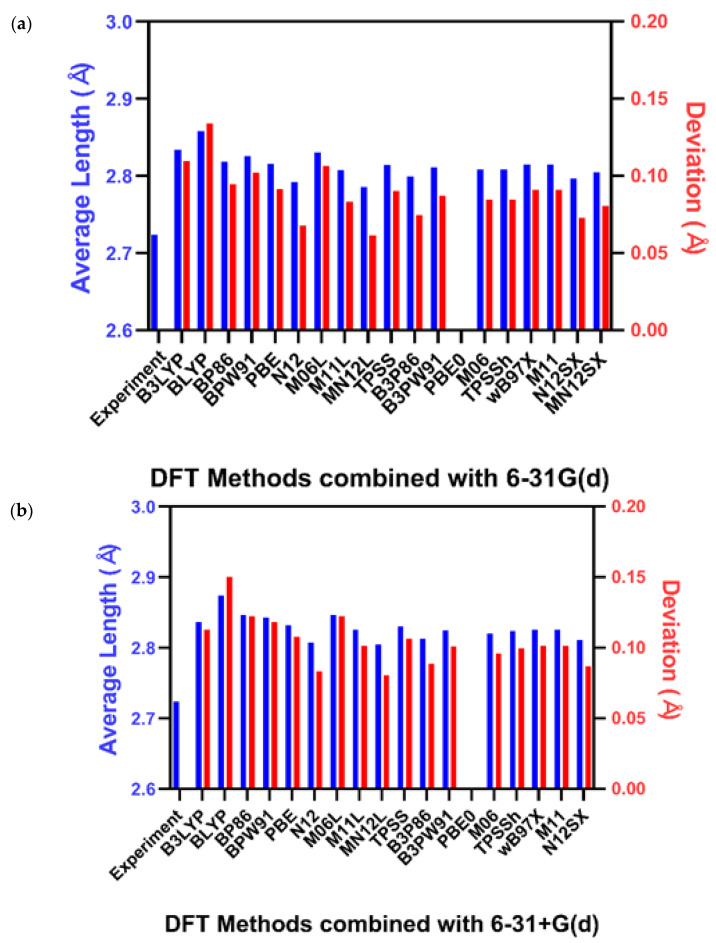
Average bond length of Am-Cl bonds in the optimized structure with (**a**) 6-31G(d) and (**b**) 6-31+G(d) [9].

**Figure 4 molecules-27-01500-f004:**
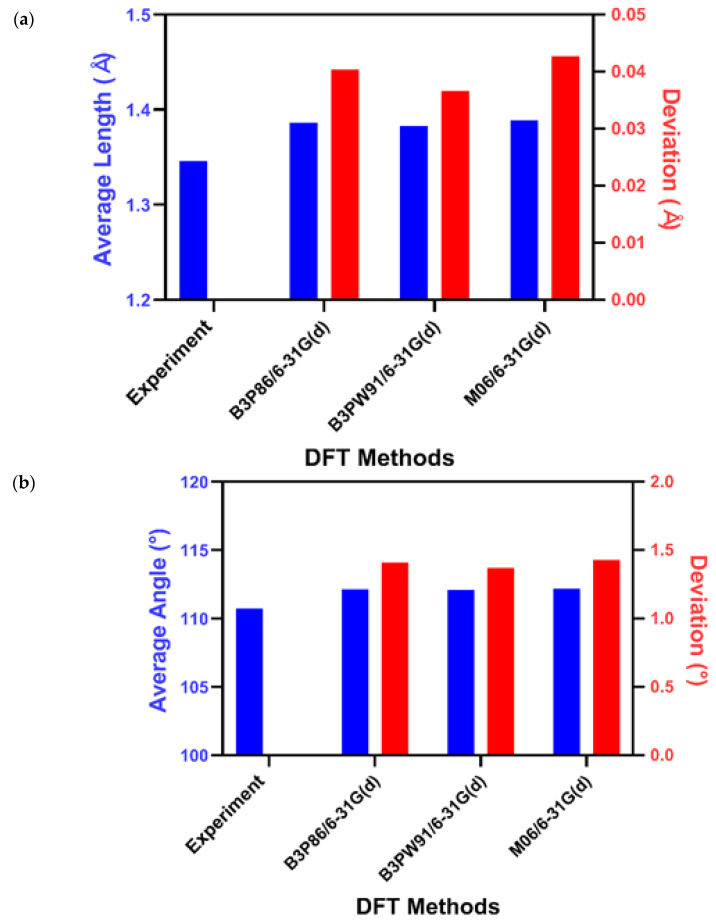
(**a**) Average bond length and (**b**) average angle of UO_2_(L)(MeOH) in the optimized structure [17].

**Table 1 molecules-27-01500-t001:** Average bond length and average angle of UO_2_(L)(MeOH) in the optimized structure.

DFT Method Combination		Average Length (Å)	Deviation (Å)	Average Angle (°)	Deviation (°)
Experiment [17]		1.34601	-	110.7458	-
6-31G(d)	B3P86	1.386322	0.040312	112.1528	1.407
	B3PW91	1.382651	0.036641	112.1132	1.3674
	M06	1.388692	0.042682	112.1715	1.4257

## Data Availability

Data is contained within the article or Appendix A.

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
