# Peer review of "Assessment of Various Density Functional Theory Methods for Finding Accurate Structures of Actinide Complexes"

_molecules, 2022, doi:10.3390/molecules27051500_

Round 1

Reviewer 1 Report

In this manuscript, the authors report an “Assessment of various density functional theory methods for 2 finding accurate structures of actinide complexes.” Without a doubt, the past 30 years have shown the usefulness of DFT computational methods to provide valuable information about the structural and electronic properties of molecules, including the reactivity and physicochemical properties of organometallic systems. However, the manuscript has some weak points that recommend a major revision before its publication.

Comments:

- The references for the DFT methods cited along the manuscript should be cited at least most of them i.e., B3LYP, BLYP, BP86, BPW91, PBE, N12, M06L, M11L, MN12L, TPSS, B3P86, B3PW91, PBE0, M06, TPSSh, wB97X, M12, N12.

- Last reviews, dealing with the assessment of DFT and their advanced applications across chemistry, physics, and materials science, have received much atention and should be considered i.e., Rev. Mod. Phys. 2015, 87, 897−923 ; Chem. Soc. Rev., 2021,50, 8470-8495; Molecular Physics, 2015, doi :10.1080/00268976.2014.1002552.

- As stated by the authors, for actinides systems, a recent review dealing with DFT Investigations of actinides systems, could be considered: Coordination Chemistry Reviews 266–267 (2014) 110–122; Magnetochemistry 2019, 5, 15; doi:10.3390/magnetochemistry5010015

- The reference cited for the Gaussian 9 software package [18], should be completed by all authors as required in the literature.

- The ref [16] should be replaced by a more relevant citation to the UF6 compound, and not a WEB citation, as done for the ref [6].

- Furthermore, the early actinide hexafluoride AnX6 species, were a subject of various DFT theoretical works, Various methodological developments are reviewed (Comput Chem 20: 70–90, 1999; J. Am. Chem. Soc. 2018, 140, 51, 17977). The authors should include them, and comment shortly on them.

- The paper (line 112-114) does not discuss clearly the results related to the additional diffuse function, notably, the increased covalent bonding in An-X bonding. The covalency in f-element systems is a subject of debate, and the uranium species are often exhibiting more pronounced covalent U-L bonding character (Nat Commun 10, 634 (2019). https://doi.org/10.1038/s41467-019-08553-y).

- English language should be revised and fixing some spelling errors and misprints.

Author Response

Dear reviewer

We thank you and the reviewers for your thoughtful suggestions and insights. The manuscript has benefited from these insightful suggestions. I look forward to working with you and the reviewers to move this manuscript closer to publication in Molecules.

The manuscript has been rechecked and the necessary changes have been made in accordance with your suggestions. The responses to all comments have been prepared and attached herewith/given below

Comments / answer:

- The references for the DFT methods cited along the manuscript should be cited at least most of them i.e., B3LYP, BLYP, BP86, BPW91, PBE, N12, M06L, M11L, MN12L, TPSS, B3P86, B3PW91, PBE0, M06, TPSSh, wB97X, M12, N12.

: We checked that all the DFT methods used in the manuscript were properly cited in the references.

- Last reviews, dealing with the assessment of DFT and their advanced applications across chemistry, physics, and materials science, have received much attention and should be considered i.e., Rev. Mod. Phys. 2015, 87, 897−923 ; Chem. Soc. Rev., 2021,50, 8470-8495; Molecular Physics, 2015, doi :10.1080/00268976.2014.1002552.

: We carefully checked your comment and added the DFT method's recent growing attention in the introduction section (p1 line 25).

- As stated by the authors, for actinides systems, a recent review dealing with DFT Investigations of actinides systems, could be considered: Coordination Chemistry Reviews 266–267 (2014) 110–122; Magnetochemistry 2019, 5, 15; doi:10.3390/magnetochemistry5010015

: We checked your opinion about the actinide system and added their comments in the introduction section (p2. Line 46).

- The reference cited for the Gaussian 9 software package [18], should be completed by all authors as required in the literature.

: We checked the software reference and revised it in reference 27.

- The ref [16] should be replaced by a more relevant citation to the UF6 compound, and not a WEB citation, as done for the ref [6].

: Instead of web data, we found and replaced reference data. You can check it in reference 23.

- Furthermore, the early actinide hexafluoride AnX6 species, were a subject of various DFT theoretical works, Various methodological developments are reviewed (Comput Chem 20: 70–90, 1999; J. Am. Chem. Soc. 2018, 140, 51, 17977). The authors should include them, and comment shortly on them.

: We reviewed the data you advised and mentioned the relevant information in the introduction. (p2 line 206 and p1 line 46)

- The paper (line 112-114) does not discuss clearly the results related to the additional diffuse function, notably, the increased covalent bonding in An-X bonding. The covalency in f-element systems is a subject of debate, and the uranium species are often exhibiting more pronounced covalent U-L bonding character (Nat Commun 10, 634 (2019). https://doi.org/10.1038/s41467-019-08553-y).

: We appreciate the reviewer’s comment. We additionally discussed the controversial structure-directing role of covalency with citation and stated our results with qualified expression. (p5 line 98)

- English language should be revised and fixing some spelling errors and misprints.

: We received professional editing from the native in this document.

Thank you for your consideration. I look forward to hearing from you.

Sincerely,

Keunhong Jeong

Reviewer 2 Report

The manuscript "Assessment of various density functional theory methods for        finding accurate structures of actinide complexes" the authors state that Density functional theory (DFT) is a widely used computational method for predicting   the physical and chemical properties of metals and organometals. DFT calculations for actinide complexes are quite demanding because their complexity increases with the increase in the number of electrons and orbitals that make up an atom. Moreover, there has been a lack of research on reasonable levels of theory for calculating the structures of actinide complexes. In this study, to find theoptimal calculation combination for predicting the geometries of actinide complexes, 38 calculations from a number of different combinations were performed on molecules containing two representative actinides. Among the 38 calculations, four optimal combinations were identified and selected for comparison with experimental data. Based on the results, the optimal combinations were applied to a more complicated and practical actinide compound, Uranyl complex (UO2(2,2′-(1E,1′E)-(2,2dimethylpropane-1,3-dyl)bis(azanylylidene)(CH3OH)) for further confirmation. As a result, the optimal calculation combination found is expected to provide a reasonable level of theory for optimizing the accurate structure of actinide complexes using DFT.

since the authors have used the theoretical calculations therefore, i encourage them to add the below sentence in the end of section 2 (Computational methods) and cite the references there in:

"The DFT method has proven to be one of the most accurate methods for the computation of the electronic structure of solids [1-7]."

[1] Journal of Molecular Structure 1248 (2022) 131462

[2] Journal of Molecular Graphics and Modelling 104 (2021) 107841, https://doi.org/10.1016/j.jmgm.2021.107841

[3] Indian J Phys (2021) https://doi.org/10.1007/s12648-020-01913-1

[4] Eur. Phys. J. Plus (2021) 136:624 https://doi.org/10.1140/epjp/s13360-021-01590-x

 therefore, i encourage the authors to revised their manuscript carefully before I give the final decision.

Author Response

Dear reviewer

We thank you and the reviewers for your thoughtful suggestions and insights. The manuscript has benefited from these insightful suggestions. I look forward to working with you and the reviewers to move this manuscript closer to publication in Molecules.

The manuscript has been rechecked and the necessary changes have been made in accordance with your suggestions. We added your comment in section3 method and references 34-37.

Thank you for your consideration. I look forward to hearing from you.

Sincerely,

Keunhong Jeong

Reviewer 3 Report

See attached file

Author Response

Dear reviewer

We thank you and the reviewers for your thoughtful suggestions and insights. The manuscript has benefited from these insightful suggestions. I look forward to working with you and the reviewers to move this manuscript closer to publication in Molecules.

The manuscript has been rechecked and the necessary changes have been made in accordance with your suggestions. The responses to all comments have been prepared and attached herewith/given below

  1. The Introduction does not provide enough background and motivation for the research. Citation of previous work in the field is very limited and should be judiciously enlarged.

: We added some content to enlarge the motivation and background in the introduction sector.

  1. Methods.
  2. The number of DFT functional employed is rather small with respect to the number of existing DFT functionals (e.g., see doi/10.1080/00268976.2017.1333644). Furthermore, the functionals were not selected on an a-priori criterion (e.g., functional yielding best structure of transition metal complexes) nor do they represent the various types of functionals (LSDA, GGA, metaGGA, hybrids, meta-hybrids, doubly hybrids, range-separated, dispersion-including, …) in an unbiased way. So, the selection seems to be small and arbitrary. It should be enlarged following the above lines.

: We understand the reviewer’s suggestion on using more various functionals, however, studied DFT functionals are not arbitrary but from previous papers, from which DFT functionals were collected and compared. As the reviewer knows, all DFT functionals do not apply to the actinide complexes and we had some experiences, which do not converge in many cases (DFT functionals). Furthermore, this study also provides a more profound result, which suggests several DFT functionals on large molecular structures and has less effective with diffusion function on basis set selection. Therefore, we concluded this series of studies is sufficient to report to the area of this field.

  1. One functional is missing from the list at p. 2 ll. 63-64.

: Thank you for your careful review. We added the methods in 3. Methods sector. (p7 line 127)

  1. Pople’s basis sets for light atoms are rather obsolete and should be replaced by better basis sets such as the Stuttgart def2 family or Jensen’s pcseg family.

: Thank you for the reviewer’s comment. The review’s perspective is right. However, much large portion of actinide-related papers (almost all papers that we cited) is using Pople’s basis set instead of quite the newest basis set. We think that is very important for us because this paper is not just comparing various methods or basis sets but providing a better method for studying actinide chemistry based on the structure information. Therefore, following the major trend was important to design this research.

  1. Finally, no mention is made of relativistic contributions to energetics and structure, which are expected to be significant for actinides.

: We understand the reviewer’s idea. Therefore, we added the comment on the relativistic effect in the manuscript as follows. (p7 line 145)

  1. Results.
  2. Throughout the paper, americium hexachloride is written as “AmCl6”. I assume that the Authors investigated [AmCl6] 3– anion. The lack of charge indication must be corrected.

: As your commented, we revised all the things that spelled chemical words.

  1. The Authors compare experimental bond lengths and angles directly to the calculated minimum-energy-structure parameters. However, it is known that these two quantities are inherently different. The experimental values are affected by vibrational effects (ZP, Boltzmann average) and packing effects. The Authors should consider and discuss these issues.

: We understand the reviewer’s worrisome. The two quantities are the major indicators for estimating the accuracy of calculation methods in the same field literature. Further, structural information is normally from x-ray crystallography and there is no way of measuring the bond length and bond angles, that is the reason what the theorists are using. That is the only way for using the experimental data.

  1. Discussion of the results is virtually absent. Which are the best functionals? How the structural parameters vary with the different functional type (LSDA, GGA, metaGGA, hybrids, meta-hybrids, doubly hybrids, range-separated, dispersionincluding, …)? What are the differences due to the inclusion of diffuse functions?

: We answered the reason for using the selected methods above. The discussion about diffuse and structural parameters is presented and described further in the discussion. Thank you for the comment for being a more understandable paper.

  1. The value  0.4 Å at p. 3 l. 95 must be corrected.

: Thank you for your careful review. We revised it in the results sector. (p3 line 80)

  1. What does the sentence “the combination of calculations with small deviations was estimated to be less than approximately 0.03 Å.” (p. 3 l. 96) means?

: As the sentence would not be meaningful in the discussion, we deleted it.

  1. Figures 1-3 contain redundant information. I suggest that each Figure contains a single panel with the deviation (calc-exp) of parameters calculated both without and with diffuse functions. The deviations for the best functionals should be tabulated in the main text; a full table should be provided in the SI.

: That is a good option for a better presentation. However, we wanted to share full tables that make up graphs and are comparable.

  1. Why are the results much poorer for [AmCl6] 3– than for UF6?

:  We appreciate the reviewer’s comment. We added the possible reason for the difference. (P5 line 94)

  1. Why was N12 used for the large complex in view of the fact that MN12L gave better results for [AmCl6] 3– ?

: Before applying to the larger uranyl complex from the small complex calculation results, the accuracy difference between N12 and MN12N for AmCl63- was less than from UF6. Further, a more accurate method for UF6 was applied for the uranyl complex.

  1. The explanation at p. 5 l. 114 is not convincing. Uranyl is a cation, so an increased covalent bonding would benefit from diffuse functions. An increased ionic character would not.

: We appreciate the reviewer’s comment. We explain more about it with additional citations and qualified expressions.

  1. Comparison of calculated and experimental average bond length and angle for the large complex is not significant because the complex is far from the octahedral structure and the bonds are very different from each other. One must compare individual bond lengths and angles. The results could be summarized in a single number by a suitably weighted “mean-of-the-mean” approach to making the comparison between functionals clearer.

: We appreciate the reviewer’s comment. We additionally analyzed the bond length and bond angle and the result is presented in the manuscript. (p5. Line 113, p6 line 117)

  1. There is no Supporting Information accompanying the paper. At least, all optimized molecular structures and their energy must be provided.

: We want to share our raw materials and tables that makeup graphs.

  1. Although the English language is good, when reading this paper, one has a feeling of general inaccuracy.

: We received professional editing from the native in this document. In addition, we thoroughly reviewed our document and revised all of the mistakes that you had commented on us.  

Thank you for your consideration. I look forward to hearing from you.

Sincerely,

Keunhong Jeong

Round 2

Reviewer 1 Report

We'd like to thank the authors for their properly revised manuscript.

Author Response

Dear reviewer

We thank you for your thoughtful suggestions and insights. 

The manuscript has been rechecked and the necessary changes have been made in accordance with your suggestions. Your comments and our responses are attached in a word file. 

Reviewer 3 Report

See attached file.

Author Response

Dear reviewer

We thank you for your thoughtful suggestions and insights. The manuscript has benefited from these insightful suggestions. I look forward to working with you and the reviewers to move this manuscript closer to publication in Molecules.

The manuscript has been rechecked and the necessary changes have been made in accordance with your suggestions. The responses to all comments have been prepared and attached in a word file.

Thank you for your consideration. I look forward to hearing from you.

Sincerely,

Keunhong Jeong
